# Smad4 restricts differentiation to promote expansion of satellite cell derived progenitors during skeletal muscle regeneration

Nicole D Paris[1], Andrew Soroka[1,2], Alanna Klose[1], Wenxuan Liu[1,3], Joe V Chakkalakal[1,4,5]*

[1]Center for Musculoskeletal Research, Department of Orthopaedics and Rehabilitation, University of Rochester Medical Center, Rochester, United States; [2]Department of Pathology, University of Rochester Medical Center, Rochester, United States; [3]Department of Biomedical Genetics, University of Rochester Medical Center, Rochester, United States; [4]Stem Cell and Regenerative Medicine Institute, University of Rochester Medical Center, Rochester, United States; [5]The Rochester Aging Research Center, University of Rochester Medical Center, Rochester, United States

*For correspondence:
joe_chakkalakal@urmc.rochester.edu

**Competing interests:** The authors declare that no competing interests exist.

**Abstract** Skeletal muscle regenerative potential declines with age, in part due to deficiencies in resident stem cells (satellite cells, SCs) and derived myogenic progenitors (MPs); however, the factors responsible for this decline remain obscure. TGFβ superfamily signaling is an inhibitor of myogenic differentiation, with elevated activity in aged skeletal muscle. Surprisingly, we find reduced expression of *Smad4*, the downstream cofactor for canonical TGFβ superfamily signaling, and the target *Id1* in aged SCs and MPs during regeneration. Specific deletion of Smad4 in adult mouse SCs led to increased propensity for terminal myogenic commitment connected to impaired proliferative potential. Furthermore, SC-specific Smad4 disruption compromised adult skeletal muscle regeneration. Finally, loss of Smad4 in aged SCs did not promote aged skeletal muscle regeneration. Therefore, SC-specific reduction of Smad4 is a feature of aged regenerating skeletal muscle and Smad4 is a critical regulator of SC and MP amplification during skeletal muscle regeneration.

## Introduction

The regenerative capacity of adult skeletal muscle is endowed in a population of *Pax7*-expressing resident stem cells called satellite cells (SCs) (*Brack et al., 2012*). Genetic studies utilizing lineage labeling, as well as cell ablation, have established that *Pax7*-expressing SCs are essential for various aspects of skeletal muscle regeneration (*Relaix and Zammit, 2012*; *Liu et al., 2015*). At homeostasis, SCs reside in a quiescent state at the interface between skeletal muscle fibers (myofibers) and the surrounding basal lamina (*Brack et al., 2012*). In response to degenerative stimuli, SCs activate and undergo proliferative expansion, providing myogenic progenitors (MPs) necessary for myofiber regeneration (*Brack et al., 2012*). Skeletal muscle regeneration requires a balance between SC/MP amplification and terminal myogenic commitment in order to efficiently form multinucleated myofibers (*Brack et al., 2012*). Moreover, there is evidence that this balance is compromised in aging skeletal muscle, where changes in SC function and their surrounding environment occur, yielding defective progenitors and stem cells (*Chakkalakal et al., 2012*; *Sousa-Victor et al., 2015*;

**eLife digest** Even in adulthood, injured muscles can repair themselves largely because they contain groups of stem cells known as satellite cells. These cells divide to produce progenitor cells that later develop, or differentiate, into new muscle fibers. However as muscles get older, this repair process becomes less effective, in part because the satellite cells do not respond as strongly to injury. It remains obscure precisely why the repair process declines with age.

A protein called TGFβ is part of a signaling pathway that prevents the muscle progenitor cells from differentiating into muscle fibers, and TGFβ signaling is overactive in older muscles. Most TGFβ signaling operates via a protein called Smad4, and Paris et al. now show that older satellite cells and progenitor cells from the muscles of old mice produce less Smad4 when they are regenerating.

Next, the gene for Smad4 was deleted specifically from the satellite cells of mice. By examining the fate of these cells, Paris et al. found that Smad4 normally maintained the population of satellite cells by preventing them from differentiating into muscle fibers too soon. This was the case when both adult and aged muscle was regenerating.

All in all, Smad4 is clearly important for directing satellite cells to regenerate properly; aged cells have less Smad4 and are less able to regenerate. Future studies are now needed to determine how disrupting Smad4 in other resident cell types may influence the regeneration of muscles in mice.

*Almada and Wagers, 2016*; *Blau et al., 2015*). Therefore, elucidating factors that regulate SC and derived MP fate is critical in order to develop interventions to combat aged skeletal muscle regenerative decline, in which these cells are lost.

Transforming growth factor beta (TGFβ) superfamily signaling is crucial for the renewal and maintenance of various tissue-specific stem cell populations (*Oshimori and Fuchs, 2012*). Superfamily ligands include TGFβs, bone morphogenetic proteins (BMPs), growth and differentiation factors (GDFs), Activins, and Myostatin. These ligands bind to transmembrane type II receptors with differing specificities. This association leads to the eventual formation of transmembrane complexes composed of type I (activin-like kinases, ALKs) and type II receptor homodimers (*Massagué, 2008*). The formation of these complexes triggers the phosphorylation of receptor SMADs (R-SMADs). The R-SMADs associated with TGFβ/Activin/Inhibin and BMP ligands are SMAD2/3 and SMAD1/5/8 respectively. Phosphorylated R-SMADs, associated with the cofactor SMAD4, accumulate in the nucleus, where together with chromatin modifiers and other transcriptional co-factors they promote the expression of target genes. Depending on cell type and context, the TGFβ superfamily pathways regulate many cellular processes including differentiation, renewal, quiescence, and apoptosis (*Oshimori and Fuchs, 2012*; *Massagué, 2012*). In aged regenerating skeletal muscle, TGFβ superfamily signaling is widely considered to be abnormally elevated, and is thought to inhibit SC activation and terminal myogenic differentiation (*Kollias and McDermott, 2008*; *Trendelenburg et al., 2012*, *2009*). However, it has also been proposed that during regeneration TGFβ superfamily ligands may enable SC and MP expansion (*Ono et al., 2011*; *Sinha et al., 2014*).

Smad4 is recognized as the canonical cofactor for TGFβ superfamily signaling and was initially identified as a tumor suppressor in pancreatic cancer (*Malkoski and Wang, 2012*; *Miyaki and Kuroki, 2003*; *Hussein et al., 2003*; *Nishita et al., 2000*). Loss-of-function *SMAD4* mutations lead to familial juvenile polyposis, which can be associated with hereditary hemorrhagic telangiectasia, and aggressive forms of various cancers (*Malkoski and Wang, 2012*; *Akhurst, 2004*). In myogenic culture systems derived from immortalized cell lines or using sequential pre-plate techniques, knockdown of Smad4 promotes myogenic differentiation (*Dey et al., 2012*; *Ono et al., 2011*). Furthermore, global reduction of expression through direct intramuscular injection of Smad4 siRNAs or viral vectors with Smad4 shRNAs into injured mouse skeletal muscle can promote the formation of larger regenerated muscle fibers relative to controls (*Dey et al., 2012*; *Lee et al., 2015*). However, given that multiple non-myogenic cell types, such as inflammatory cells and fibro/adipogenic progenitors, also contribute to SC and MP fate decisions during skeletal muscle regeneration; It is unclear which cellular mechanisms promote hypertrophy of regenerated myofibers with non-targeted Smad4 loss. In contrast, specific loss of Smad4 in MPs compromises myogenic differentiation

during embryonic skeletal muscle development (*Han et al., 2012*). Additionally, consistent with the critical role for Smad4 in stem and progenitor cell function, targeted deletion of Smad4 in hematopoietic, hair follicle, and neural stem and derived progenitor cell populations leads to their depletion during homeostasis and regeneration (*Karlsson et al., 2007*; *Yang et al., 2009*; *Mira et al., 2010*). Moreover, targeted loss of Smad4 in myofibers leads to modest deterioration during growth and aggravation of denervation-induced atrophy in adults (*Sartori et al., 2013*). Recently, gain-of-function *SMAD4* mutations that prevent ubiquitination and subsequent degradation have been identified as the cause of the rare developmental disorder Myhre syndrome in humans (*Caputo et al., 2014*; *Le Goff et al., 2012*). Patients with Myhre syndrome are characterized by short stature, various musculoskeletal abnormalities, and hypertrophied musculature (*Caputo et al., 2014*; *Le Goff et al., 2012*). Although Smad4 clearly has crucial roles in skeletal muscle and tissue-specific stem and progenitor cell biology, to date no studies have explicitly examined whether or not there is a cell-autonomous requirement for Smad4 in SCs and derived MPs during skeletal muscle regeneration.

In this study we show, in comparison to adult, evidence of failure to induce *Smad4* expression in aged SCs and MPs during skeletal muscle regeneration. In order to examine the consequences of cell-specific Smad4 loss, we utilized transgenic mice expressing tamoxifen-inducible Cre recombinase under the control of *Pax7* regulatory elements to perform targeted deletion of Smad4 in SCs. We found that specific disruption of Smad4 in adult SCs resulted in insufficient SC and derived MP amplification, which was accompanied by severe deficiencies in adult skeletal muscle regeneration. Unexpectedly, with specific loss of Smad4 in aged SCs in an environment of presumably high TGFβ activity, aged skeletal muscle regeneration was not improved.

## Results

### Smad4 expression is reduced in aged SCs and myogenic cells during regeneration

Deficiencies in aged skeletal muscle regeneration reflect in part a failure or delay of SC or SC-derived MP expansion due to multiple factors. These factors include impaired activation, premature terminal fate commitment, and the occurrence of senescence and apoptosis (*Sousa-Victor et al., 2015*). Since SMAD-dependent signaling and target genes such as *Id1* have been implicated in the regulation of the terminal fate and amplification of SC and MP populations (*Ono et al., 2011*; *2012*; *Clever et al., 2010*), we examined the expression of *Smad4* and *Id1* in SCs and MPs from regenerating adult and aged skeletal muscles. Initially, we employed previously characterized flow cytometric analysis to examine age-related modification of SMAD4 protein levels in SCs and MPs (Lin-, Sca1-, ITGA7+) isolated from adult and aged, uninjured and regenerating skeletal muscle. Regenerating muscle was examined at five days post injury (5dpi), a time point when new myofibers are rapidly forming through the expansion, differentiation, and fusion of SC-derived myogenic cells (*Murphy et al., 2011*; *Cosgrove et al., 2014*; *Bernet et al., 2014*; *García-Prat et al., 2016*). To induce skeletal muscle regeneration, a barium chloride ($BaCl_2$) solution was directly injected into tibialis anterior (TA) muscles, which is an established model of skeletal muscle degeneration and regeneration (*Murphy et al., 2011*). Relative to SCs from adult uninjured TAs, an approximately 2.5-fold increase in SMAD4 protein was observed in SCs and MPs isolated from adult 5dpi TA muscles (*Figure 1A and C*). In contrast, SMAD4 induction was not detected in SCs and MPs isolated from aged 5dpi relative to uninjured TA muscles (*Figure 1B and C*). To further substantiate these findings, we conducted RTqPCR analysis of *Smad4* expression as well as the SMAD-target *Id1,* the loss of which is associated with deficiencies in skeletal muscle regeneration (*Clever et al., 2010*). Both *Smad4* and *Id1* expression were higher in SCs and MPs from adult when compared to those from aged 5dpi TAs (*Figure 1D and E*). We were unable to obtain consistent Ct values in the detectable range (<36) for *Smad4* or *Id1* in SCs from uninjured skeletal muscle indicative of low to negligible expression (data not shown). Therefore, a feature of age-related regenerative decline is the loss of Smad4 induction in SCs and MPs.

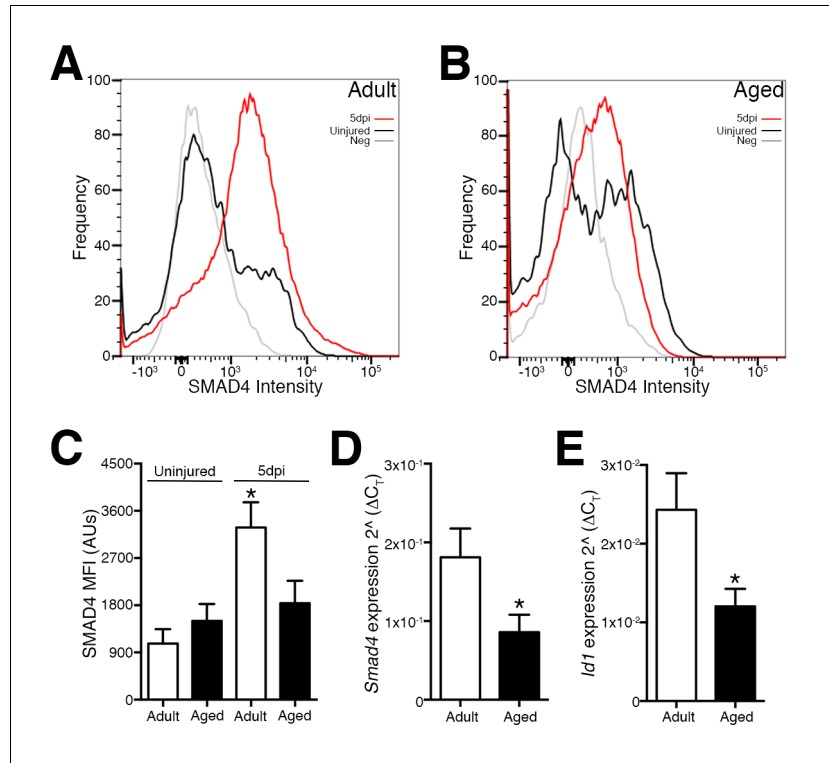

**Figure 1.** Loss of Smad4 expression in aged satellite cells and myogenic progenitors during muscle regeneration. Representative profiles of SMAD4 protein levels by flow cytometric analysis in SCs and MPs isolated from C57BL6 (**A**) adult and (**B**) aged uninjured and 5-day post injured (5dpi) (TA) muscles. (**C**) Quantification of SMAD4 protein levels in adult and aged SCs and MPs from uninjured and 5dpi TA muscles. (**D**) Quantification of *Smad4* expression in FACs-sorted SCs and MPs isolated from 5dpi adult and aged TA muscles. (**E**) Quantification of SMAD target *Id1* expression in FACs-sorted SCs and MPs isolated from 5dpi adult and aged TA muscles. N = 4 mice, for (**C**) *p<0.05 ANOVA and Fisher's test, (**D**) and (**E**) *p<0.05 t-test.

## Smad4 loss severely impairs SC clonal growth and proliferative potential

To begin investigating the consequences of Smad4 loss in adult SCs, *Smad4*[flox/flox] mice were bred with mice expressing a tamoxifen-inducible Cre under the control of *Pax7* regulatory elements (*Pax7*[+/CreERT]), enabling satellite cell-specific Smad4 disruption. To characterize Smad4 loss in *Pax7*[+/CreERT]; *Smad4*[flox/flox] (P7:S4KO) and *Pax7*[+/+]; *Smad4*[flox/flox] (Ctl) animals, adult mice were administered tamoxifen (Tmx), SCs and MPs from 5dpi TAs were prospectively isolated, and the expression of *Smad4* and the SMAD-target *Id1* were interrogated (*Figure 2—figure supplement 1*). As a validation of successful deletion, flow cytometric analysis revealed that SMAD4 protein induction fails to occur in P7:S4KO SCs from 5dpi relative to uninjured skeletal muscles in comparison to the induction seen in Ctl muscles (*Figures 2A,B and C*). As expected, we also observed a reduction in *Smad4* and *Id1* mRNA levels in P7:S4KO when compared to Ctl SCs and MPs from 5dpi TAs (*Figure 2D and E*). We have also previously demonstrated a very high efficiency of *Pax7*[CreERT] recombination in adult SCs (*Liu et al., 2015*). Therefore, both Smad4 expression and function are efficiently lost in P7:S4KO SCs and MPs.

Previous studies have shown that knockdown of Smad4 or TGFβ superfamily ligand supplementation in myogenic cells is able to promote or impair terminal myogenic commitment, respectively (*Ono et al., 2011*; *Dey et al., 2012*; *Lee et al., 2015*; *Trendelenburg et al., 2009*). To examine the terminal myogenic fate decisions of Smad4-deficient SCs, FACs-purified adult P7:S4KO and Ctl SCs and MPs were cultured for 96 hr in plating media (10% horse serum, FGF2, DMEM), and supplemented with vehicle, TGFβ1, or BMP4 ligands (*Chakkalakal et al., 2012*, *2014*; *Zammit et al., 2004*). Based on immunofluorescence analysis, all cells regardless of genotype or supplementation

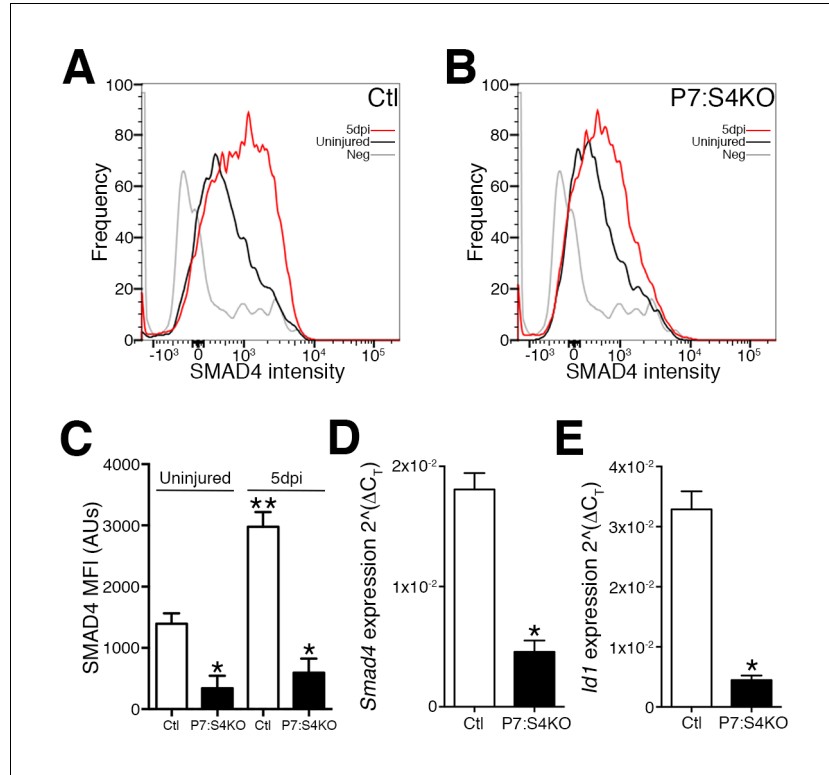

**Figure 2.** Disruption of Smad4 expression in P7:S4KO satellite cells and myogenic progenitors during muscle regeneration. Representative profiles of SMAD4 protein levels by flow cytometric analysis in SCs and MPs isolated from adult (A) Ctl and (B) P7:S4KO uninjured and 5dpi TA muscles. (C) Quantification of SMAD4 protein levels by flow cytometry of Ctl and P7:S4KO SCs and MPs from uninjured and 5dpi TA muscles. (D) Quantification of *Smad4* expression in FACs-sorted SCs and MPs isolated from 5dpi Ctl and P7:S4KO TA muscles. (E) Quantification of SMAD target *Id1* mRNA levels in FACs-sorted SCs and MPs isolated from 5dpi Ctl and P7:S4KO TA muscles. N = 4 mice, for (C) *p<0.05 to Ctl, **p<0.05 Adult Ctl ANOVA and Fisher's test, for (D) and (E) *p<0.05 t-test.

The following figure supplement is available for figure 2:

**Figure supplement 1.** Gating of control and P7:S4KO SCs for intracellular protein analysis.

were labeled by either Pax7 (SC renewal marker) or Myogenin (terminal myogenic commitment marker) (*Figure 3A*). Consistent with previous reports, both TGFβ1 and BMP4 supplementation led to a lower proportion of Ctl SC-derived cells that were Myogenin-positive (*Figure 3C*) (*Ono et al., 2011*). Notably, cultures derived from P7:S4KO SCs contained a significantly lower proportion of Pax7+ cells; However, we observed a higher proportion of Myogenin+ cells compared to Ctl, regardless of supplementation (*Figure 3B and C*). We subsequently tested whether the enhanced terminal myogenic commitment observed in P7:S4KO cultures was associated with greater myogenic fusion and formation of immature multinucleated muscle fibers (myotubes). To achieve this, SCs were cultured at high density (10,000 cells/well) to ensure adequate numbers of fusion-competent MPs. Cultures were then immunostained with pan anti-skeletal muscle myosin antibody to visualize myotubes (*Figure 3D*). In accordance with an increased propensity to differentiate, quantification of nuclei within skeletal muscle myosin-expressing cells (fusion index) revealed this parameter to be ~two fold greater in P7:S4KO compared to Ctl myotubes (*Figure 3E*).

Specific Smad4 deletion in embryonic tongue MPs leads to impaired myogenic differentiation attributed to reduced *Fgfr4* and *Fgf6* expression (*Han et al., 2012*). Given that we observed enhanced terminal myogenic commitment in culture, we then examined the expression levels of pertinent FGF ligands and receptors in five day cultures from Smad4-deleted adult SCs and derived MPs. Although we observed reductions in *Fgfr4* and *Fgf6* expression, we also identified a decrease

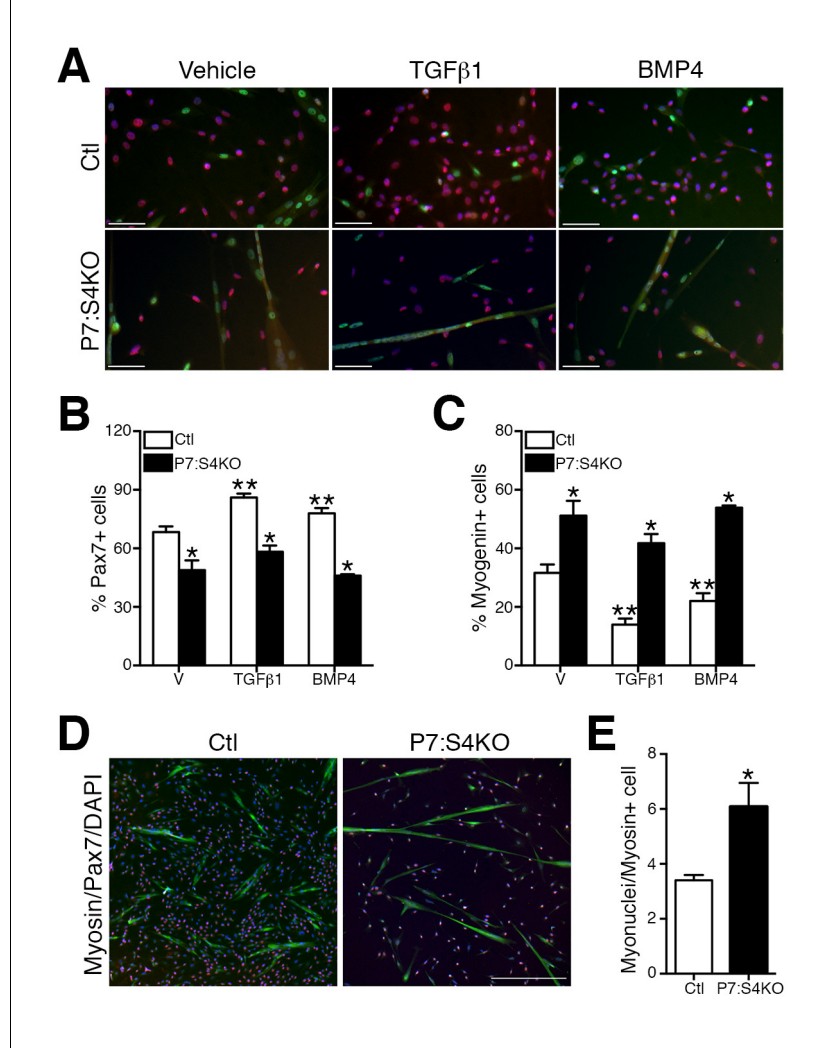

**Figure 3.** Enhanced terminal myogenic commitment in P7:S4KO satellite cells and myogenic progenitors. (**A**) Representative images of Pax7 (red), Myogenin (green), and DAPI (blue) immunofluorescence in FACs-sorted adult Ctl or P7:S4KO 96 hr SC cultures (plated at 4000 cells/well) treated with vehicle, TGFβ1, or BMP4. (**B**) Quantification of Pax7 immunofluorescence in Ctl and P7:S4KO FACs-sorted SC cultures. (**C**) Quantification of Myogenin immunofluorescence in Ctl and P7:S4KO FACs sorted SC cultures. (**D**) Representative images of Pax7 (red), skeletal muscle myosin (green), and DAPI (blue) immunofluorescence in adult FACs-sorted Ctl or P7:S4KO myotube cultures (plated at 10000 cells/well). (**E**) Quantification of fusion index; myonuclei/myosin+ cell. N = 3 cultures, For (**B**) and (**C**) *p<0.05 significant to Ctl, **p<0.05 significant to vehicle Ctl, ANOVA Fishers test, for (**E**) *p<0.05 t-test, scale = 50 μm.

The following figure supplement is available for figure 3:

**Figure supplement 1.** Modulations in FGFR and FGF expression following SC-specific Smad4 loss in adult mice.

in *Fgfr1* and an induction of *Fgf2* expression (*Figure 3—figure supplement 1*). Therefore, unlike what is seen in embryonic MPs from tongue muscle, loss of *Fgf6* and *Fgfr4* expression as a result of specific Smad4 disruption in adult SCs is associated with increased terminal commitment, reduced *Fgfr1* and elevated *Fgf2* expression.

Next, we assessed the proliferative potential of Ctl and P7:S4KO SC cultures. FACs-purified adult P7:S4KO and Ctl SCs and MPs were cultured in plating media for 72 hr, and pulsed with the thymidine analog EdU for the last 4 hr (*Sousa-Victor et al., 2014*). The vast majority of cells in 72 hr cultures, regardless of genotype, were labeled by the myogenic fate markers Pax7 or MyoD

(*Figures 4A,C and D*). In comparison to Ctl, the proportion of P7:S4KO cells that had incorporated EdU was significantly reduced (*Figure 4B*). We performed further immunofluorescence analysis of MyoD and Pax7 labeling to assess the fate of Ctl and P7:S4KO SCs in 72 hr cultures (*Figure 4C*). Although no difference was observed in the proportion of cells that were Pax7+/MyoD-, P7:S4KO cultures displayed higher percentages of Pax7-/MyoD+ cells and a lower proportion of Pax7+/MyoD

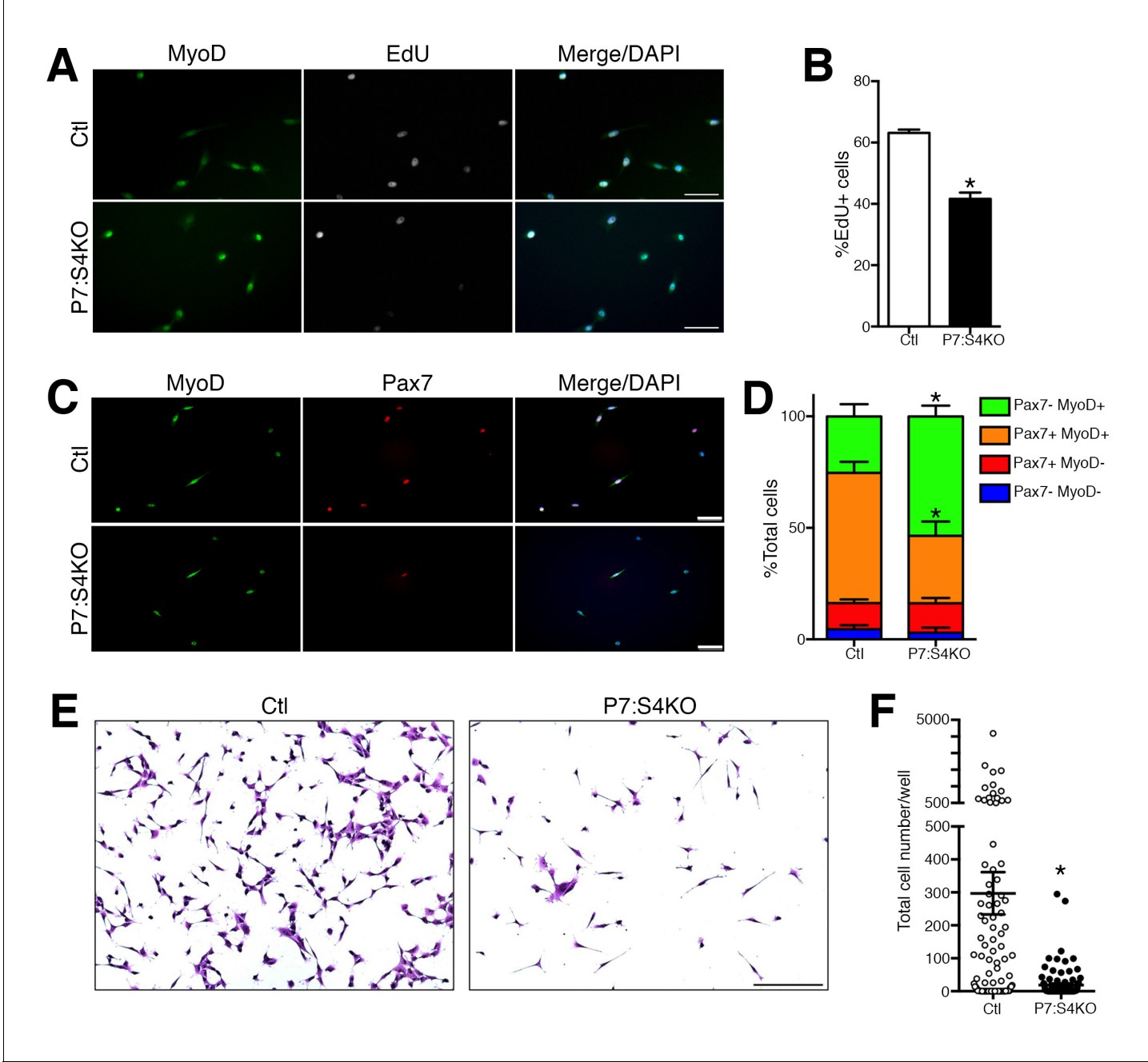

**Figure 4.** Reduced proliferation in P7:S4KO satellite cells and myogenic progenitors. (**A**) Representative images of MyoD (green), EdU (grey), and DAPI (blue) immunofluorescence of FACs-sorted adult Ctl or P7:S4KO 72 hr SC cultures pulsed with EdU for the last 4 hr. (**B**) Quantification of the proportion of cells that are EdU+. N = 3 cultures, *p<0.05 t-test,scale = 50 μm. (**C**) Representative images of MyoD (green), Pax7 (red), and DAPI (blue) immunofluorescence of FACs-sorted adult Ctl or P7:S4KO 72 hr SC cultures, scale = 50 μm. (**D**) Quantification of the proportion of cells that are Pax7+ and/or MyoD+. N = 3 cultures, *p<0.05 t-test. (**E**) Representative images of Crystal Violet-stained 7-day cultures of FACs-sorted adult Ctl or P7:S4KO myogenic cells plated at clonal density (10 cells/well), scale = 200 μm. (**F**) Quantification of cell growth. N = 96 cultures, *p<0.05 t-test.

+ cells (*Figure 4D*). Collectively, these observations provide additional evidence that specific disruption of Smad4 in SCs promotes terminal myogenic commitment, which is associated with reduced proliferative potential.

To further assess these cellular mechanisms, we sought to determine the consequences of Smad4 loss on SC clonal growth potential. FACs-purified adult Ctl and P7:S4KO SCs were sorted directly into 96 well plates at 10 cells per well, cultured for seven days, and stained with Crystal Violet to enable counting with conventional light microscopy (*Figure 4E*). Consistent with our observation of reduced proliferative potential, we found the clonal growth potential of P7:S4KO SCs to be significantly reduced compared to Ctl (*Figure 4F*).

## Loss of Smad4 in adult SCs impairs progenitor expansion during regeneration

To initially examine SC fate in vivo, adult P7:S4KO and Ctl 5dpi TA muscles were processed to assess Pax7+ SC proliferation and number. To identify proliferating Pax7+ SCs at 5dpi, a pulse of the thymidine analog BrdU was i.p. administered at 2 hr prior to sacrifice. Quantification of the proportion of BrdU+/Pax7+ cells revealed a significant reduction in proliferating as well as total Pax7+ SCs in 5dpi P7:S4KO muscles (*Figure 5A–C*). Additionally, we examined the expression of cell cycle inhibitors *Cdkn1a* (p21), *Cdkn1b* (p27), and *Cdkn2a* (p16). No induction in the expression of cell cycle inhibitors was observed in adult P7:S4KO SCs and MPs isolated by FACs from 5dpi TAs (*Figure 5—figure supplement 1*). Furthermore, no significant difference was observed between P7:S4KO and Ctl SC number in uninjured TAs 21 days after Tmx treatment (*Figure 5—figure supplement 2*). Therefore, the loss of SCs in P7:S4KO 5dpi TAs does not reflect initial declines in Pax7+ SCs upon recombination at homeostasis.

To examine MP number and fate, adult Ctl and P7:S4KO 5dpi muscle sections were processed for the detection of MyoD and active Caspase-3 (aCasp, apoptosis marker) by immunofluorescence (*Figure 6A*). Consistent with the declines seen in Pax7+ SC proliferation and number, we found that MyoD+ cell number was reduced in P7:S4KO 5dpi muscles (*Figure 6C*). Although we observed a significant reduction in MyoD+ cells, the proportion of MPs that were aCasp+ was similar in both Ctl and P7:S4KO 5dpi muscles (*Figure 6D*). Therefore, SC-specific Smad4 disruption leads to impaired SC and MP amplification that does not coincide with heightened cell death.

To determine if reduced SC and MP amplification upon Smad4 loss coincides with an increased propensity for terminal commitment in vivo, adult Ctl and P7:S4KO 5dpi muscle sections were processed for the immunofluorescent detection of Pax7 and MyoD together (Pax7+MyoD) and Myogenin (MyoG), recognized with separate fluorescent-conjugated secondary antibodies (*Figure 6B*). The examination of Pax7/MyoD+ cells with Myogenin labeling enables the detection of total myogenic cells and based on the proportion that are Myogenin+, the extent of terminal myogenic commitment can be determined. Consistent with our SC-derived culture data, we found that a significantly higher proportion of myogenic cells were terminally committed in P7:S4KO 5dpi muscles (*Figure 6E*). Therefore, reduced SC and MP amplification resulting from Smad4 loss coincides with an increased drive toward terminal myogenic commitment during muscle regeneration.

## SC-specific Smad4 loss severely impairs skeletal muscle regeneration at all ages

To assess the consequences of Smad4 deletion on proper muscle regeneration, we examined the size of Ctl and P7:S4KO 5dpi regenerated myofibers identified with embryonic Myosin Heavy Chain (eMyHC, marker of recent regeneration) and Laminin staining by immunofluorescence (*Figure 7A*). In agreement with what would be expected given the impairment in SC-derived MP amplification, the size (cross-sectional area) of eMyHC-positive myofibers was reduced in adult 5dpi regenerated skeletal muscles following SC-specific Smad4 disruption (*Figure 7C*).

In aged regenerating skeletal muscle, elevated TGFβ activity is thought to impair SC and myogenic progenitor amplification (*Carlson et al., 2008*; *Pessina et al., 2015*). Furthermore, non-specific knockdown of Smad4 through siRNAs or viral delivery of shRNAs to regenerating aged skeletal muscle promotes myofiber hypertrophy (*Dey et al., 2012*; *Lee et al., 2015*). Therefore, we sought to examine whether specific disruption of Smad4 in aged SCs could promote aged skeletal muscle regeneration, presumably in an environment of high TGFβ ligand activity, suggesting that the

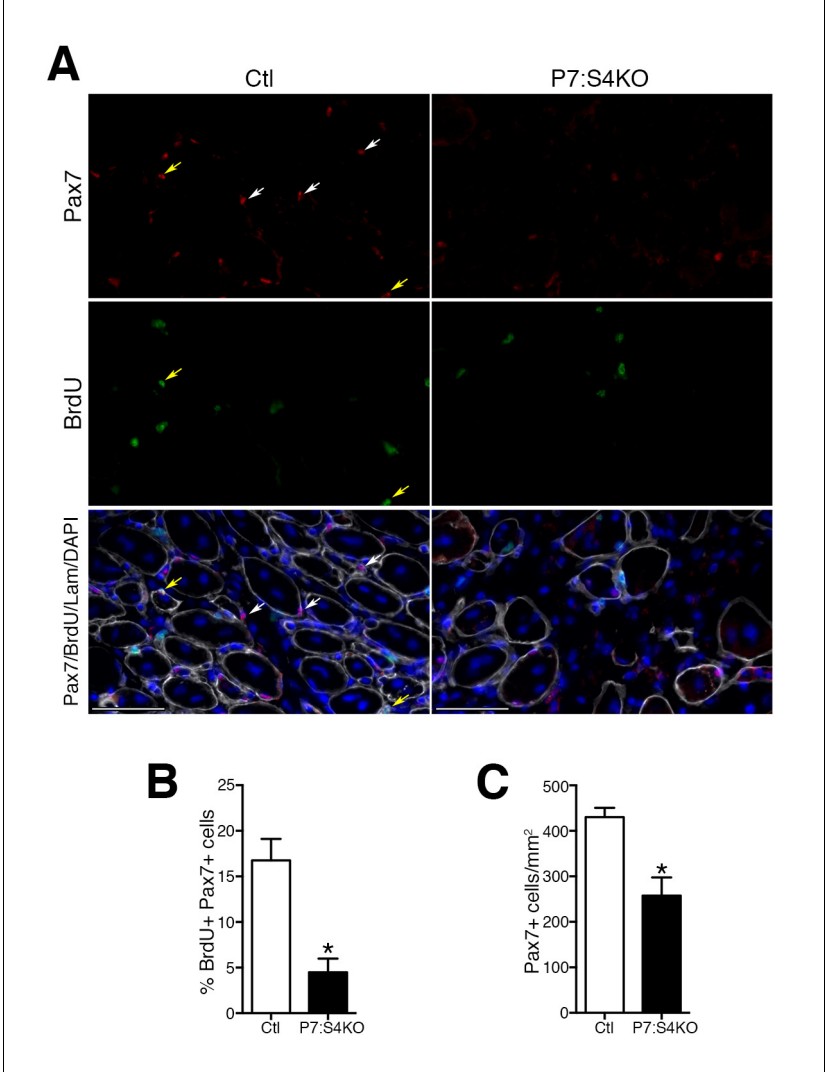

**Figure 5.** Smad4 disruption induces loss of proliferating and total satellite cell number during muscle regeneration. (**A**) Representative images of Pax7 (red), BrdU (green), DAPI (blue) and Laminin (grey) immunofluorescence in adult 5dpi Ctl and P7:S4KO TA muscle sections. (**B**) Quantification of the proportion of BrdU+ Pax7+ cells in adult 5dpi Ctl and P7:S4KO TA muscle sections. (**C**) Quantification of Pax7+ cells in adult 5dpi Ctl and P7:S4KO TA muscle sections. N = 4 mice, *p<0.05 t-test, scale = 50 μm. Pax7+ cells (white arrows) and Pax7+BrdU+ (yellow arrows).

The following figure supplements are available for figure 5:

**Figure supplement 1.** Smad4 disruption does not induce *Cdkn1a*, *Cdkn1b*, or *Cdkn2a* expression in SCs and MPs sorted from adult regenerating TA muscle.

**Figure supplement 2.** Smad4 disruption does not induce loss of Pax7+ SCs in uninjured TA muscles.

implications of signaling loss could differ from results observed in adult mice. To test this hypothesis, P7:S4KO and Ctl mice were aged to 22 months and treated with Tmx. Thereafter, a TA muscle was degenerated with intramuscular injection of BaCl$_2$, and subsequently allowed to regenerate for five days. In accordance with Smad4 disruption in adult mice, expression of *Smad4* and the SMAD-target *Id1* was reduced in FACs-isolated SCs and MPs from aged P7:S4KO 5dpi TAs, validating efficient and similar Cre-mediated Smad4 deletion at both ages (*Figure 7—figure supplement 1A and B*). As described elsewhere, aged SCs displayed elevated expression of *Cdkn2a* (p16), however the

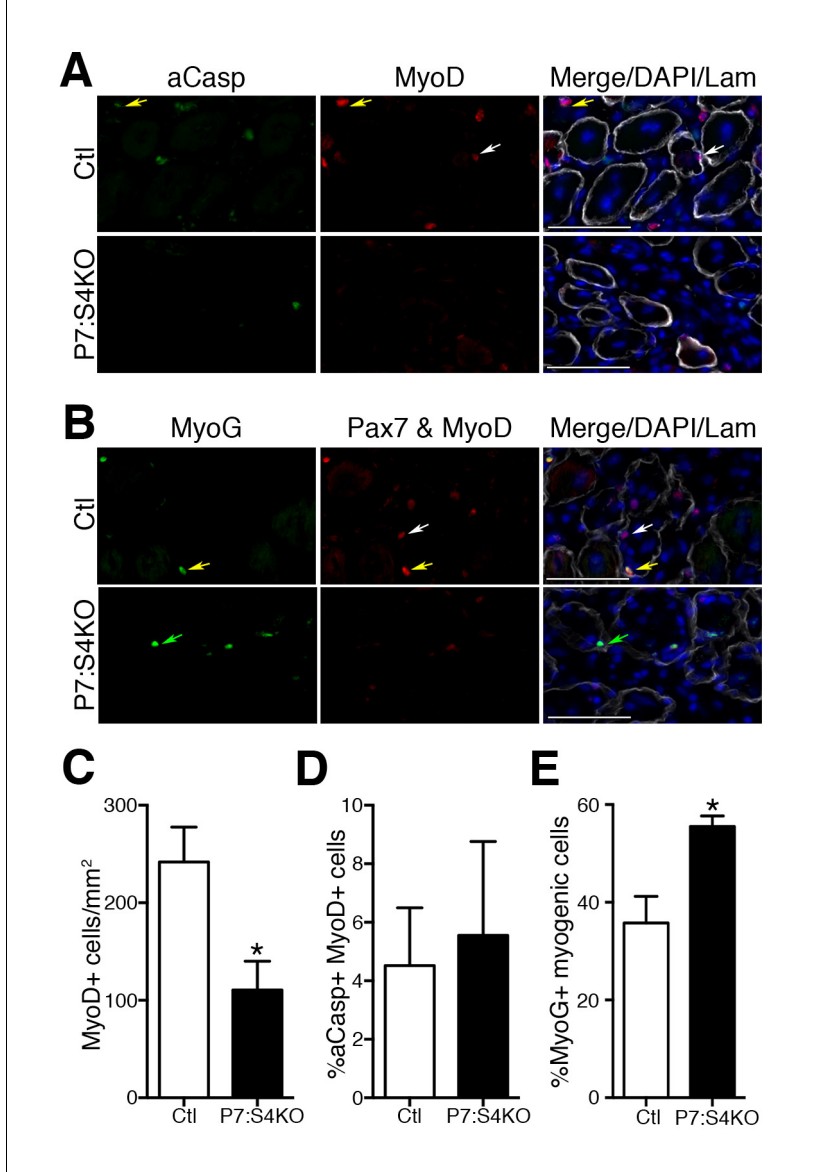

**Figure 6.** Smad4 disruption enhances terminal myogenic commitment during muscle regeneration. (**A**) Representative images of active Caspase 3 (aCasp, green), MyoD (red), DAPI (blue), and Laminin (grey) immunofluorescence of adult 5dpi Ctl and P7:S4KO TA muscle sections. MyoD+ cells (white arrows) and MyoD +aCasp+ (yellow arrows). (**B**) Representative images of Myogenin (MyoG, green), Pax7 and MyoD (red), DAPI (blue) and Laminin (grey) immunofluorescence of adult 5dpi Ctl and P7:S4KO TA muscle sections. (**C**) Quantification of MyoD+ cells in 5dpi Ctl and P7:S4KO TA muscle sections. (**D**) Quantification of the proportion of aCasp+ MyoD+ cells in 5dpi Ctl and P7:S4KO TA muscle sections. (**E**) Quantification of the proportion of MyoG+ Pax7 and MyoD+ cells in 5dpi Ctl and P7:S4KO TA muscles. Pax7 and MyoD+ cells (white arrows) and Pax7 and MyoD+ MyoG+ (yellow arrows) MyoG+ (green arrows). N = 4 mice, *p<0.05 t-test, scale = 50 μm.

extent of increased expression did not differ in response to Smad4 disruption (*Figure 7—figure supplement 1C*) (*Sousa-Victor et al., 2014*; *Bernet et al., 2014*; *Cosgrove et al., 2014*; *Carlson et al., 2008*). Based on quantification of regenerated eMyHC-positive myofiber size, specific loss of Smad4 in aged SCs did not promote aged regeneration (*Figure 7B and C*).

To determine if persisting regenerative deficits could be detected in aged and adult P7:S4KO mice, additional TA muscles were analyzed at 14 days following BaCl$_2$ injury, a timepoint at which regeneration should essentially be complete. Regardless of genotype, uninjured TAs demonstrated

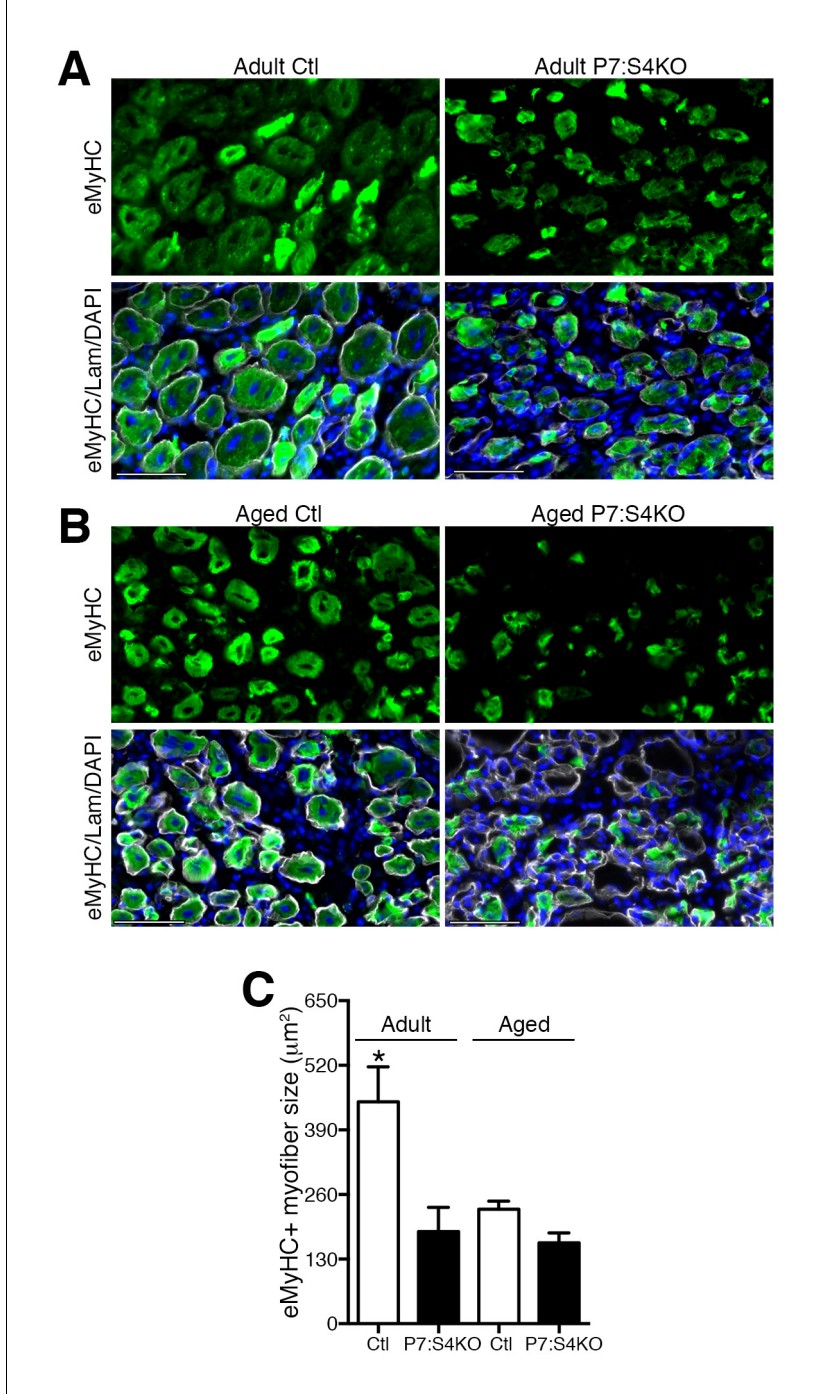

**Figure 7.** Smad4 disruption in satellite cells impairs skeletal muscle regeneration. Representative images of embryonic Myosin Heavy Chain (eMyHC, green), DAPI (blue) and Laminin (grey) immunofluorescence of 5dpi Ctl and P7:S4KO (A) adult and (B) aged TA muscle sections. (C) Quantification of average eMyHC+ regenerated myofiber size in 5dpi adult and aged Ctl and P7:S4KO TA muscles. N = 4 mice, 250–300 myofibers. *p<0.05 ANOVA Fishers test, scale = 50 μm.

The following figure supplement is available for figure 7:

**Figure supplement 1.** Smad4 disruption reduces *Smad4* and the SMAD target *Id1* expression in adult and aged SCs and MPs sorted from regenerating TA muscle.

similar age-related myofiber atrophy (*Figure 8A,B and C*). Examination of 14dpi TAs revealed that, at all ages, insufficient muscle regeneration persisted following SC-specific Smad4 loss (*Figure 8B and D*). Collectively, these data identify Smad4 as a critical muscle stem cell regulator, maintaining the appropriate balance between SC-derived MP amplification and terminal commitment during aged and adult skeletal muscle regeneration.

## Discussion

Here we find Smad4 to be a factor that is normally induced in adult, but lost in aged, SCs and MPs in regenerating muscles. Furthermore, we demonstrate that in an inducible mouse model driving specific loss of Smad4 in SCs, a decline in the number of proliferating Pax7+ SCs and MPs, and consequently severe deficits in skeletal muscle regeneration are experienced regardless of age. Several studies have specifically manipulated components of TGFβ superfamily signaling in myogenic progenitors and SCs (*Guardiola et al., 2012*; *Han et al., 2012*; *Huang et al., 2014*). However, until now, no reports have examined the cell-autonomous roles of Smad4, the common cofactor for all

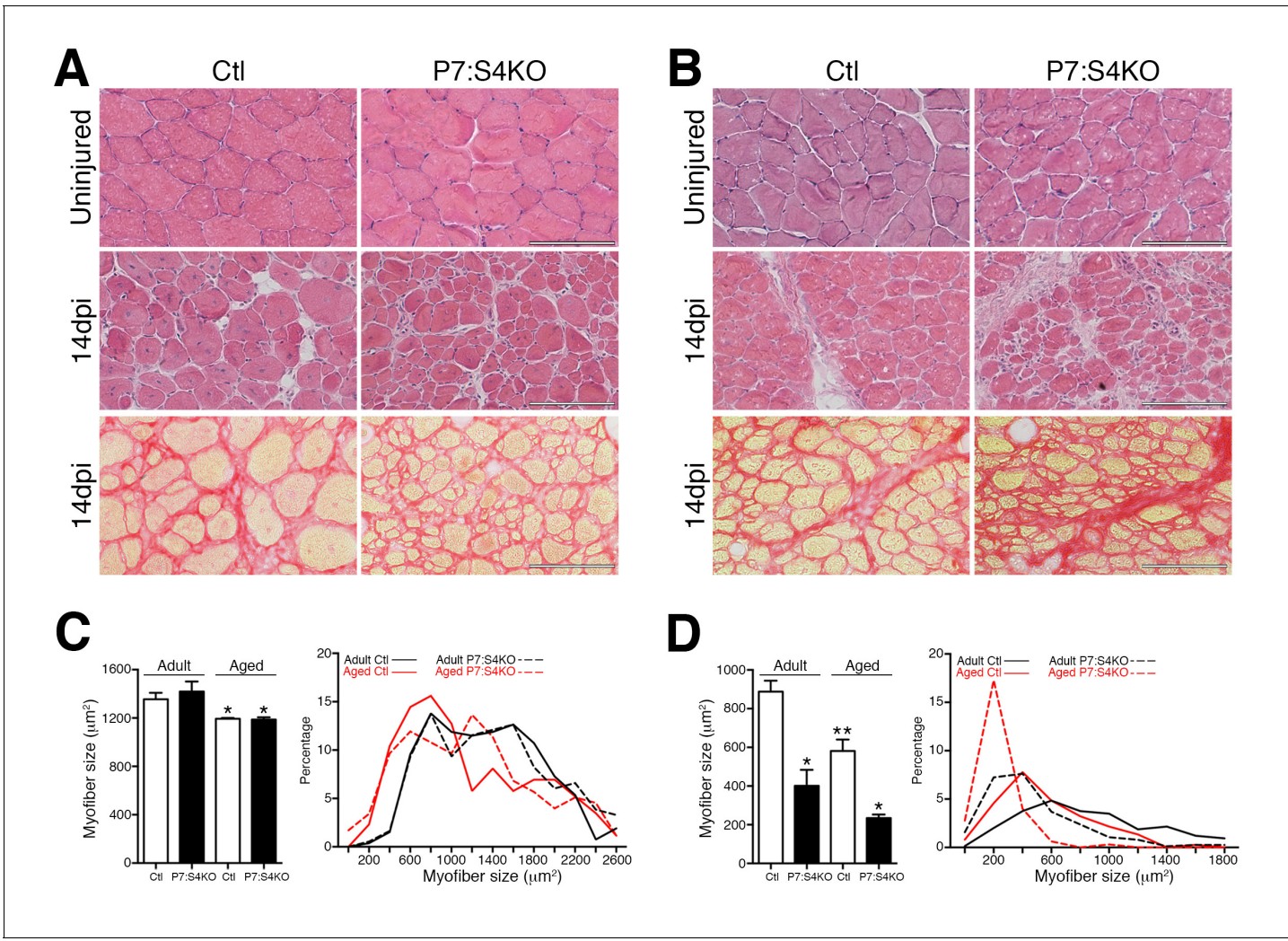

**Figure 8.** Smad4 disruption in satellite cells leads to persistent deficits in skeletal muscle regeneration. Representative images of H and E-stained uninjured and 14dpi, and Sirius Red-stained 14dpi (**A**) adult and (**B**) aged TA muscle sections. Quantification and frequency distribution of (**C**) uninjured and (**D**) 14dpi myofiber size in adult and aged Ctl and P7:S4KO TA muscles. N = 4 mice, 700–900 myofibers. For (**C**) *p<0.05 to Adult, for (**D**) *p<0.05 to Ctl, **p<0.05 to Adult Ctl, ANOVA Fisher's test, scale = 100 μm.

branches of canonical TGFβ superfamily signaling, in adult or aged Pax7+ SCs and determined the subsequent consequences on skeletal muscle regeneration.

Consistent with previous reports, Smad4 disruption did promote myogenic terminal commitment (*Dey et al., 2012*; *Ono et al., 2011*; *Lee et al., 2015*). However, declines in proliferating SCs and MPs during regeneration were also observed. Myofibers are multinucleated cells formed by the fusion of many terminally committed myogenic cells. Therefore, the progression to the formation of mature multinucleated myofibers can be impeded at multiple levels, including impaired myogenic differentiation and/or reduced MP amplification through a variety of mechanisms: inhibition of cell cycle entry, cell death, and/or premature terminal commitment (*Sousa-Victor et al., 2015*; *Brack et al., 2012*). Given that we have detected differences in myogenic cell proliferation with no increased cell death or cell cycle inhibitor expression, we suspect that the reduced number of MPs in regenerating skeletal muscles following acute SC-specific Smad4 loss is likely due to a heightened propensity for terminal commitment. In contrast, specific disruption of Smad4 in embryonic mouse MPs, driven by *Myf5-Cre*, altered terminal myogenic commitment and myofiber formation in the tongue (*Han et al., 2012*). Intuitively, these divergent results could reflect differences in the intrinsic mechanisms that regulate SCs and MPs in response to the environments of embryonic tongue growth versus adult TA regeneration. Another possibility is that the *Myf5-Cre*, in addition to targeting the myogenic lineage, drives recombination in further resident cell populations with the capacity to directly or indirectly influence skeletal muscle growth and regeneration (*Huang et al., 2014*). It will be of interest to determine in what manner specific disruption of Smad4 in other resident non-myogenic cell populations may influence adult or aged skeletal muscle regeneration. Although in adult Smad4 null SC cultures we observed reductions in *Fgf6* and *Fgfr4* expression similar to embryonic Smad4 deleted muscle, we also detected a robust induction of *Fgf2*. We have previously found that the presence of elevated levels of FGF2 in aged skeletal muscle is associated with a propensity for aged SCs to progress toward terminal fates (*Chakkalakal et al., 2012*). Therefore, compensatory induction of FGF2 resulting from Smad4 loss may be a unique feature of adult SCs that drives terminal myogenic commitment, the mechanisms of which will require further investigation.

Previous work has shown that loss of BMP signaling induces premature terminal myogenic commitment, preventing SC and MP amplification (*Ono et al., 2011*). Furthermore, Id1, which we found to be reduced in aged wildtype and adult P7:S4KO SCs and derived MPs, was identified as a SMAD target required for SC and MP amplification (*Ono et al., 2011*). Indeed, through direct interactions Id1 can also promote Myogenin degradation (*Vinals and Ventura, 2004*). Therefore, loss of Id1 in aged and P7:S4KO SCs and MPs during regeneration could potentially stabilize Myogenin and thus promote excessive terminal commitment that impedes myogenic cell amplification. Although we observe loss of *Smad4* and *Id1* expression, some reports have described the presence of elevated TGFβ and phospho-Smad activity in aged SCs and MPs from regenerating skeletal muscle (*Carlson et al., 2008*; *Pessina et al., 2015*). Although less characterized, TGFβ can function through non-canonical Smad4-independent pathways (*Derynck and Zhang, 2003*; *Massagué, 2012*). For instance, pSmad3, which is elevated in aged SCs and MPs, also associates with Drosha in a Smad4-independent complex required for microRNA processing (*Davis et al., 2008*). Studies examining keratinocyte differentiation have shown Iκβ kinase to be a critical regulator of signaling through Smad2/3 that is Smad4-independent (*Descargues et al., 2008*). Non-canonical TGFβ mediators such as TAK1 (TGFβ activated kinase 1) and the downstream target p38 are both shown to be elevated in aged myogenic cells (*Cosgrove et al., 2014*; *Bernet et al., 2014*; *Trendelenburg et al., 2012*). Notably, heightened p38 activity in aged SCs hinders myogenic cell amplification through mechanisms that include promotion of terminal myogenic commitment (*Cosgrove et al., 2014*; *Bernet et al., 2014*). Collectively, it will be critical to determine if loss of Smad4 in aged and P7: S4KO SCs and MPs could impair myogenic cell amplification through promotion of multiple Smad4-independent pathways.

Although the TGFβ superfamily pathways are capable of influencing multiple cell types that contribute to the regeneration of skeletal muscle, few studies have examined specific inhibition of these pathways in adult or aged SCs and their derived progenitors. For example, it has been shown that SC-specific loss of Cripto, an inhibitor of Activin, Myostatin, and TGFβ signaling in myogenic cells, leads to smaller regenerated myofibers (*Guardiola et al., 2012*). Additionally, genetic ablation of the BMP receptor *Alk3* in embryonic myogenic cells, utilizing *MyoD-Cre* and *Myf5-Cre* mice, was found to result in ineffective adult skeletal muscle regeneration, however this occurs primarily

through mechanisms other than major dysfunction within the SC pool (*Huang et al., 2014*). This conclusion was based primarily on the lack of observable depletion or dysfunction of *Alk3* null SCs in vivo or in culture, respectively, which is thought to occur due to stimulation of the expression of the related *Alk6* as a compensatory mechanism to maintain SC function (*Huang et al., 2014*). In this study, we determined that inducible *Pax7*-driven loss of Smad4 impedes SC-derived myogenic proliferation and proper skeletal muscle regeneration. Further studies will be needed to determine in what way targeted disruption of Alk3, induced later in life, in adult or aged SCs may affect skeletal muscle regeneration at these ages, with potential compensation possibly being eliminated.

Although not specifically targeting SCs or MPs, intramuscular delivery of Smad4 siRNA or shRNA at later time points during regeneration has been shown to stimulate myofiber hypertrophy (*Dey et al., 2012*; *Lee et al., 2015*). Furthermore, utilizing miR-26 disruption to increase, and miR-431 mimics to decrease, *Smad4* expression at later timeponts during regeneration promotes myofiber atrophy and hypertrophy respectively (*Dey et al., 2012*; *Lee et al., 2015*). However, it is intuitive that since microRNAs often target many genes, the aforementioned manipulations likely do not alter *Smad4* expression alone. Therefore, it remains unclear whether the balance between myogenic progenitor expansion and terminal fate commitment could be manipulated to be beneficial to adult and aged muscle if cell-specific Smad4 loss was induced at different timepoints during regeneration or by alternative, more transient, means.

Here, we demonstrate that induction of Smad4 is compromised in aged SCs and myogenic cells during skeletal muscle regeneration. Disruption of Smad4 specifically in adult SCs leads to phenotypes observed in aged regenerating skeletal muscle including compromised proliferation and amplification of SC-derived MPs. Collectively, these data indicate that Smad4 function in SCs is essential for adult and aged skeletal muscle regeneration in the mouse. Moving forward, careful dissection of Smad4-mediated pathways and targets may yield novel factors that promote skeletal muscle genesis in the contexts of aging and disease.

## Materials and methods

### Animals

All procedures involving mice were carried out in accordance with guidelines set by the Animal Care and Use Committee at the University of Rochester. Adult (3–6 months) and aged (22–24 months) mice were housed in the animal facility with free access to standard rodent chow and water. All mouse strains were obtained from Jackson Laboratory (Bar Harbor, ME): C57BL/6J, *Pax7CreER* (017763), and *Smad4flox/flox* (017462). Aged C57BL6 mice were obtained from the National Institute on Aging. PCR genotyping was performed using protocols described by Quanta, with primer sequences and annealing temps provided by JAX. To induce Cre recombination, mice were injected i.p. with 100 µL of 20 mg/mL (~60 µg/kg) tamoxifen (Tmx, Sigma-Aldrich, St. Louis, MO, T5648 in 90% Sigma corn oil/10% EtOH) for five consecutive days, with clearance for five days prior to injury. To examine Pax7+ SC numbers at homeostasis, muscles were harvested 21 days after Tmx administration. Tmx-injected *Pax7+/+;Smad4flox/flox* littermates in each cohort were used as controls.

### Skeletal muscle injury

Mice were anesthetized with an i.p. injection of ketamine (100 mg/kg) and xylazine (10 mg/kg) or by 1–3% isoflurane inhalation. Buprenorphine (0.1 mg/kg) was administered prior to the procedure and approximately every 12 hr as needed. The skin overlaying the tibialis anterior (TA) muscle was shaved and the TA was directly injected with a 1.2% solution of $BaCl_2$ in normal saline. At five days post-injury, injured and contralateral (uninjured) TAs were collected. TAs collected for immunofluorescence were incubated at 4°C overnight in 30% sucrose prior to embedding in OCT and flash freezing.

### Skeletal muscle primary cell isolation

To obtain highly purified SCs and MPs, primary cells were isolated from regenerating and uninjured muscles as described previously (*Chakkalakal et al., 2012*; *Pessina et al., 2015*). Hindlimb muscles were isolated and myofiber fragments were obtained by Type II Collagenase (Gibco, Carlsbad, CA) digestion, trituration, and multiple sedimentation. Mononucleated cells were liberated by further

Type II Collagenase and Dispase (Gibco) digestion, trituration, sedimentation and filtration. Cells were stained with CD31, Sca1, CD45 (BD Biosciences, San Jose, CA, #561410, #562058, Biolegend, San Diego, CA, #103132), and Integrin α7 (AbLab, Vancouver, Canada, clone R2F2) fluorescent-conjugated antibodies. Cells were collected using a FACSAria II Cell Sorter (BD Biosciences). Live SCs/MPs were isolated using forward and side scatter profiles, negative selection for DAPI, CD31/45 and Sca1, and positive selection for Integrin α7.

## Intracellular flow cytometry

Injured and contralateral uninjured TAs were digested as described above. SCs/MPs were identified by an alternate cell surface staining panel selecting for Integrin α7 and negatively selecting for CD31, CD45, and Sca1 (Biolegend #102420, #103132, #108127). Cell fixation, permeabilization, and intracellular staining of Smad4 (Santa Cruz Biotechnology, Santa Cruz, CA, sc-7966 PE) using BD Fixation Buffer (#554655) and BD Phosflow Perm Buffer III (#558050) was carried out according to the BD Phosflow protocol. Analysis of cell surface and intracellular staining was performed on an LSR II Flow Cytometer (BD Biosciences). Mean Fluorescent Intensity (MFI) was quantified with FlowJo software. Negative controls stained with all cell surface markers but not intracellular antibodies were analyzed to assess background staining. Negative control MFI was subtracted to obtain final MFI values.

## FACs-purified SC/MP cultures

FACs-purified SCs/MPs were plated at 4000 cells per well in eight-well Permanox chamber slides (Nunc, Rochester, NY) and cultured for five days in plating media (10% Horse Serum, 5 ng/mL FGF2, DMEM). The ligands TGFβ1 (10 ng/mL) and BMP4 (10 ng/mL), as well as DMSO vehicle (1 μL/mL), were added to cultures beginning on day 3. Cultures were immunostained for Pax7 and Myogenin to characterize P7:S4KO and Ctl terminal myogenic commitment. To assess myotube formation, FACs-purified SCs/MPs were plated at 10,000 cells per well and cultured for five days in plating media, ensuring sufficient cell density to promote fusion. Myotube cultures were immunostained for skeletal muscle myosin and DAPI to calculate fusion index (myonuclei per myosin-positive cell). To assess clonal cell growth, FACs-sorted SCs were plated at clonal density (10 cells per well in 96 well plates) and the number of Crystal Violet-stained cells present in each individual well was determined after seven days in culture.

## Cell proliferation analysis

In SC/MP cultures (plated at 5000 cells per well), proliferative capacity was assayed by EdU (5-ethynyl-2′-deoxyuridine) incorporation using the Click-iT EdU Alexa Fluor 647 Imaging Kit (Molecular Probes, Carlsbad, CA, C10640). Cells were grown in basal media for 72 hr and incubated with EdU for the last 4 hr. EdU detection was performed according to manufacturer protocol and was followed by immunostaining for Pax7 and MyoD. In vivo cell proliferation in Ctl and P7:S4KO mice was assayed by immunostaining for Pax7, BrdU (5-bromo-2′-deoxyuridine), and Laminin after 250 μL of 2 mg/ml (~15 μg/kg) BrdU (Sigma) was i.p. injected two hours prior to harvest of injured and contralateral uninjured TA muscles.

## Immunofluorescence

Dissected TA muscles were incubated overnight at 4°C in 30% sucrose, flash frozen, cryosectioned at 10 μm, and stored at −80°C prior to staining. Muscle sections were fixed for 3 min in 4% paraformaldehyde (PFA), and if needed, subjected to antigen retrieval: incubation in citrate buffer (10 mM sodium citrate, pH 6.0) in a steamer (Oster #5712) for 15 min after 10 min preheating of buffer (*Liu et al., 2015*; *Tang et al., 2007*). Tissue sections were permeabilized with PBS-T (0.2% Triton X-100) for 10 min and blocked in 10% Normal Goat Serum (NGS; Jackson Immuno Research, West Grove, PA) in PBS-T for 30 min at room temperature. When mouse primary antibodies were used, sections were additionally blocked in 3% AffiniPure Fab fragment goat anti-mouse IgG(H+L) (Jackson Immuno Research) with 2% NGS in PBS at room temperature for 1 hr. Primary antibody incubation in 2% NGS/PBS was carried out at 4°C overnight or 2 hr at RT and sections were incubated with secondary antibodies in 2% NGS/PBS for 1 hr at RT. DAPI staining was used to label nuclei. All slides were mounted with Fluoromount-G (SouthernBiotech, Birmingham, AL). At least four sections from

three slides were analyzed per sample. Immunocytochemistry was performed following the same protocol with the exception of the Fab blocking step. Sections and cells were imaged on a Zeiss Axio Observer A.1 microscope (Germany).

## Skeletal muscle regeneration assay

Sections of TA muscles were harvested at five or 14 days post-injury from adult or aged Ctl and P7:S4KO mice. Regenerated skeletal muscles harvested five days after injury were immunostained with embryonic Myosin Heavy Chain (eMyHC) antibodies to label actively regenerating fibers and average cross-sectional area of 250–300 myofibers was quantified with ImageJ. Regenerated skeletal muscles harvested 14 days after injury were processed for H and E or Sirius Red staining. For H and E, flash-frozen sections were fixed for 3 min in 4% PFA, stained with Mayers Hematoxylin and Alcoholic Eosin Y, dehydrated, equilibrated with xylene and mounted using Cytoseal 60 (Richard-Allan Scientific, Kalamazoo, MI). For Sirius Red staining, a Picrosirius Red stain kit (Polysciences, Warrington, PA) was utilized. Frozen sections were fixed for 1 hr at 56°C in Bouin's fixative, washed in water, stained for 1 hr in Picrosirius Red, washed in 1 M HCl, dehydrated, equilibrated and mounted. Bright-field images were collected with a Zeiss Axioskop 40 microscope. To obtain quantification of average cross-sectional area and frequency distribution of 14dpi regenerated fiber size, Myosin/Laminin immunostained TA sections were analyzed using ImageJ software.

## RNA isolation and RT-qPCR

RNA was isolated from sorted SCs/MPs, or sorted SC/MP 5-day cultures for FGF analysis using phase separation in Trizol (Invitrogen, Carlsbad, CA) followed by cleanup with the RNeasy Plus Mini-kit (Qiagen, Germany), according to manufacturer protocols. To prepare sorted cell RNA for RT-qPCR, first-strand complementary DNA was synthesized from ~50 ng of RNA using the SuperScript First-Strand cDNA Synthesis Kit (Invitrogen). RT-qPCR was performed on a Step One Plus Real Time PCR machine (Applied Biosystems, Carlsbad, CA) using Platinum SYBR Green qPCR SuperMix-UDG with ROX master mix (Invitrogen). Experiments were standardized to *Gapdh*. All reactions for RT-qPCR were performed using the following thermal cycler conditions: 50°C for 2 min, 95°C for 2 min, 40 cycles of a two-step reaction, denaturation at 95°C for 15 s, annealing at 60°C for 30 s. The following primers were used:

| Primer name | Forward sequence | Reverse sequence |
| --- | --- | --- |
| Gapdh | AGGTCGGTGTGAACGGATTTG | TGTAGACCATGTAGTTGAGGTC |
| Smad4 | ACACCAACAAGTAACGATGCC | GCAAAGGTTTCACTTTCCCCA |
| Id1 | CCTAGCTGTTCGCTGAAGGC | CTCCGACAGACCAAGTACCAC |
| Cdkn1a (p21) | TCGCTGTCTTGCACTCTGGTGT | CCAATCTGCGCTTGGAGTGATAG |
| Cdkn1b (p27) | TCAAACGTGAGAGTGTCTAAC | CCGGGCCGAAGAGATTTCTG |
| Cdkn2a (p16) | CGCAGGTTCTTGGTCACTGT | TGTTCACGAAAGCCAGAGCG |
| Fgf1 | CCCTGACCGAGAGGTTCAAC | GTCCCTTGTCCCATCCACG |
| Fgf2 | GCGACCCACACGTCAAACTA | TCCCTTGATAGACACAACTCCTC |
| Fgf6 | CAGGCTCTCGTCTTCTTAGGC | AATAGCCGCTTTCCCAATTCA |
| Fgfr1 | GCCTCACATTCAGTGGCTGAAG | AGCACCTCCATTTCCTTGTCGG |
| Fgfr4 | TCCGACAAGGATTTGGCAGACC | TGGCGGCACATTCCACAATCAC |

## Antibodies

The following antibodies were used: Mouse anti-Pax7 (1:100, Developmental Studies Hybridoma Bank (DSHB), Iowa City, IA), rabbit or mouse anti-MyoD (1:250, Santa Cruz, sc-304 or BD Biosciences #554130), rabbit anti-Myogenin (1:250, Santa Cruz, sc-576), mouse anti-embryonic Myosin Heavy Chain BF-45/F1.652 (1:40, Developmental Studies Hybridoma Bank (DSHB), Iowa City, IA), rat anti-BrdU (1:250, Abcam, Cambridge, UK, ab6326), rabbit anti-Cleaved Caspase 3 (1:400, Cell Signaling, Beverly, MA, #9664), rat or rabbit anti-Laminin (1:1000 or 1:1500, Sigma-Aldrich, L0663 or

L9393), rabbit anti-skeletal muscle myosin (1:250, Sigma-Aldrich HPA1239), AlexaFluor 594-conjugated goat anti-mouse IgG (1:1500, Life Technologies, Carlsbad, CA, A-11032), AlexaFluor 488-conjugated goat anti-mouse IgG (1:1500, Life Technologies, A-11001), AlexaFluor 488-conjugated goat anti-rabbit IgG (1:1500, Life Technologies, A-11034), AlexaFluor 488-conjugated goat anti-rat IgG (1:1500, Life Technologies, A-11006), AlexaFluor 647-conjugated goat anti-rat IgG (1:1500, Life Technologies, A-21247), AlexaFluor 647-conjugated goat anti-rabbit IgG (1:1500, Life Technologies, A-21244).

## Data analysis

Immunofluorescent images were analyzed using ImageJ software. Results are presented as mean + SEM. Statistical significance was determined by Student's $t$-tests for simple comparison or by one-way ANOVA and Bonferroni multiple comparisons test for multiple comparisons with Graph Pad Prism software. $p < 0.05$ was considered statistically significant.

## Acknowledgements

We thank Terry Wightman and Matthew Cochran (University of Rochester Medical Center Flow Cytometry Core) as well as the Center for Musculoskeletal Research Histology, Biochemistry, and Molecular Imaging Core for technical assistance. This work was supported by URMC startup funds and NIH grant (R01AG051456) to JVC.

## Additional information

### Funding

| Funder | Grant reference number | Author |
| --- | --- | --- |
| University of Rochester Medical Center | Start-up Funds | Joe V Chakkalakal |
| National Institutes of Health | R01AG051456 | Joe V Chakkalakal |

The funders had no role in study design, data collection and interpretation, or the decision to submit the work for publication.

### Author contributions

NDP, Conception and design, Acquisition of data, Analysis and interpretation of data, Drafting or revising the article; AS, Designed and performed myogenic differentiation experiment and wrote some portions of the manuscript; AK, Performed all RNA isolations and RT-qPCR experiments, Analysis and interpretation of data; WL, Designed and performed assessment of SC number experiments, Analysis and interpretation of data; JVC, Conception and design, Analysis and interpretation of data, Drafting or revising the article

### Author ORCIDs

Nicole D Paris, http://orcid.org/0000-0003-0654-0983
Joe V Chakkalakal, http://orcid.org/0000-0002-8440-7312

### Ethics

Animal experimentation: This study was performed in strict accordance with the recommendations in the Guide for the Care and Use of Laboratory Animals of the National Institutes of Health. Work with mice was conducted in accordance with protocols approved by the University Committee on Animal Resources, University of Rochester Medical Center (Protocol #101565/2013-002).

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
