## [Decision Letter]

Thank you for submitting your article "Smad4 restricts differentiation to promote expansion of satellite cell derived progenitors during muscle regeneration" for consideration by *eLife*. Your article has been favorably evaluated by Sean Morrison (Senior Editor) and three reviewers, including Amy J Wagers (Reviewer #1) - who is a member of our Board of Reviewing Editors – and Colin Crist (Reviewer #2).

The reviewers have discussed the reviews with one another and the Reviewing Editor has drafted this decision to help you prepare a revised submission.

Summary:

The manuscript by Chakkalakal et al. investigates the impact of deletion of SMAD4, a common signal transducer of the TGFβ superfamily, in adult and aged mouse muscle satellite cells. The authors show that SMAD4 is normally induced during muscle repair in young adults, and furthermore, using well-established lineage-specific, inducible CreER-LoxP deletion strategies, that deletion of Smad4 accelerates satellite differentiation, reduces proliferation, and impairs muscle regeneration after injury. Interestingly, they also examine SMAD4 ablation in aged animals, where it has been suggested that elevated TGFβ signaling constrains robust muscle regenerative potential. Unexpectedly, the authors report impaired SMAD4 induction in aged, regenerating muscle and show that SMAD4 deletion in this context is unable to reverse age-related regenerative defects. These results implicate SMAD4 as a positive regulator of muscle repair, and should prompt additional dissection of the upstream signals that modulate SMAD4 levels during muscle regeneration and aging.

The overall design of this study is straightforward, and should be of interest as it raises new complexities with respect to the interactions of TGF signals and muscle repair. The authors' use of highly specific genetic systems and aging of their animals out to 2 years is an important contribution. Furthermore, the results nicely highlight important differences in the generative/regenerative responses of developing, adult and aged muscle, and make an important point about the context specificity of SMAD4 requirement and the challenges of predicting pathway activity or impact based solely on ligand levels. However, a major discussion point among the reviewers was the extent to which the work provides new insights into SMAD4 functions, especially as this is a well-studied target in development and skeletal muscle biology, and the extent to which the authors are able to resolve the apparent differences they report between a loss of Smad4 during development, which seems to prevent efficient differentiation but not proliferation, and during adulthood, which reduces proliferation presumably by promoting differentiation. There were also some technical concerns related to the sufficiency with which the authors documented recombination efficiency in their mice, whether Smad4 fails to be upregulated, or if it is already upregulated in satellite cells of aged muscle (the way the data is presented, this is not clear), and whether the defects in P7S4-/- mice persist at later time points after injury. Addressing these points will be essential to shore up the authors' conclusions, and should be achievable by this team with their existing genetic tools and expertise. If these issues are addressed appropriately, then the work will be appropriate for *eLife*.

Essential revisions:

1) Figure 1 – basal levels of SMAD4 protein (in uninjured muscle) seem higher in at least a subset of aged SCs vs. adult SCs. If this is correct it is difficult to interpret the data in Figure 1, which expresses data relative to uninjured samples. Thus, for Figure 1, quantification and statistics for multiple experiments should be included for comparison of both uninjured and 5dpi to the negative control (not just UI to 5dpi, as currently done). A similar analysis is needed for Figure 2. Smad4 and Id1 mRNA expression are likewise not reported for uninjured controls present in Figure 1, and should be.

2) The authors need to better assess the SMAD4 deficiency phenotype in satellite cells using additional markers. Pax7 expression alone does not indicate satellite cells undergoing self-renewal. Authors should quantify numbers of Pax7-positive, MyoD-negative verus Pax7-positive, MyoD-positive cells to determine if satellite cells that have not activated the myogenic program are perturbed in their ex vivo culture system. In addition, they should include additional differentiation markers (e.g. myosin heavy chain) to quantify myotube formation and fusion index to support their claim that Smad4 mutant muscle stem cells possess an enhanced propensity for differentiation.

3) The authors should provide a better evaluation of the efficiency of SMAD4 deletion in the TAM-treated P7S4-/- satellite cells in adult vs. aged mice to verify that deletion efficiency is equivalent at the two ages. If the efficiency of Smad4 gene deletion is low in either circumstance, then it would be necessary to use a recombination sensitive reporter to analyze only cells in which the cre-recombinase was successfully activated, as inclusion of all cells (recombined and not recombined) in the analysis might comprise the results (again, depending on the efficiency of recombination).

4) In Figure 8, the authors convincingly demonstrate a reduction in regenerating fiber size at 5dpi in P7S4-/- muscle and aged muscle. Do these defects persist at later timepoints? The authors should examine fiber CSA in more fully regenerated muscle (e.g., 14 and 21 dpi) and in uninjured muscle (at the same post-TAM time point as for the injury/repair studies) as well, to support their conclusions that Smad4-deletion in satellite cells results in severe deficiencies in skeletal muscle regeneration. Both HE and trichrome staining should be shown together with the distribution of fiber sizes. They could also ask if numbers of Pax7-positive cells present underneath the basal lamina return to normal levels in the absence of Smad4, although this might not be testable if they continue to observe a collapse in regeneration at the later timepoint.

5) The authors need to provide further insight into mechanism to support their conclusion that satellite cell expansion is disrupted with SMAD4 deletion. Inactivation of Smad4 during embryonic development leads to major defects in tongue muscle formation, which was attributed to impaired myogenic differentiation and fusion probably due to decreased expression of FGF4 and FGF6. Thus, the authors should investigate whether the adult phenotype similarly intersects with FGF signaling. To do so, they can use the existing data on tongue muscle development in the NCBI repository, then perform a targeted qPCR panel to identify the suspect FGF factors. The authors can add the suspect FGF exogenously to wt and Smad4 KO satellite cell cultures and ask if the defects they have observed are rescued. This will clarify whether Smad4 targets different pathways during development and adult life, which is a central tenet of the work.

---

## [Author Response]

*[…] Essential revisions:*

*1) Figure 1 – basal levels of SMAD4 protein (in uninjured muscle) seem higher in at least a subset of aged SCs vs. adult SCs. If this is correct it is difficult to interpret the data in Figure 1, which expresses data relative to uninjured samples. Thus, for Figure 1, quantification and statistics for multiple experiments should be included for comparison of both uninjured and 5dpi to the negative control (not just UI to 5dpi, as currently done). A similar analysis is needed for Figure 2. Smad4 and Id1 mRNA expression are likewise not reported for uninjured controls present in Figure 1, and should be.*

As requested Figure 1 and Figure 2 have been modified to include uninjured and 5dpi data. Negative values are subtracted from UI and 5dpi values. Smad4 and Id1 mRNA expression were examined in SCs derived from uninjured adult and aged skeletal muscles. We were unable to obtain consistent Ct values in the detectable range (< 36) indicative of extremely low/undetectable levels. This is written in the text of the Results (subsection “Smad4 expression is reduced in aged SCs and myogenic cells during regeneration”).

*2) The authors need to better assess the SMAD4 deficiency phenotype in satellite cells using additional markers. Pax7 expression alone does not indicate satellite cells undergoing self-renewal. Authors should quantify numbers of Pax7-positive, MyoD-negative verus Pax7-positive, MyoD-positive cells to determine if satellite cells that have not activated the myogenic program are perturbed in their* ex vivo *culture system. In addition, they should include additional differentiation markers (e.g. myosin heavy chain) to quantify myotube formation and fusion index to support their claim that Smad4 mutant muscle stem cells possess an enhanced propensity for differentiation.*

As requested we have conducted assessment of Pax7 and MyoD in our ex vivo culture system, the results are found in Figure 4, Results (subsection “Smad4 loss severely impairs SC clonal growth and proliferative potential”, fourth paragraph). As requested we have conducted myotube formation assays, the results are found in Figure 3, Results (subsection “Smad4 loss severely impairs SC clonal growth and proliferative potential”, second paragraph.

*3) The authors should provide a better evaluation of the efficiency of SMAD4 deletion in the TAM-treated P7S4-/- satellite cells in adult vs. aged mice to verify that deletion efficiency is equivalent at the two ages. If the efficiency of Smad4 gene deletion is low in either circumstance, then it would be necessary to use a recombination sensitive reporter to analyze only cells in which the cre-recombinase was successfully activated, as inclusion of all cells (recombined and not recombined) in the analysis might comprise the results (again, depending on the efficiency of recombination).*

As requested we have examined SMAD4 disruption in P7S4-/- satellite cells, the results are in Figure 8—figure supplement 1. These data indicate similar disruption of SMAD4 in P7S4-/- aged and adult SCs.

*4) In Figure 8, the authors convincingly demonstrate a reduction in regenerating fiber size at 5dpi in P7S4-/- muscle and aged muscle. Do these defects persist at later timepoints? The authors should examine fiber CSA in more fully regenerated muscle (e.g., 14 and 21 dpi) and in uninjured muscle (at the same post-TAM time point as for the injury/repair studies) as well, to support their conclusions that Smad4-deletion in satellite cells results in severe deficiencies in skeletal muscle regeneration. Both HE and trichrome staining should be shown together with the distribution of fiber sizes. They could also ask if numbers of Pax7-positive cells present underneath the basal lamina return to normal levels in the absence of Smad4, although this might not be testable if they continue to observe a collapse in regeneration at the later timepoint.*

As requested we have conducted experiments to examine regeneration at a later time point. Assessment of H & E and Sirius Red stained sections demonstrate SC specific Smad4 deletion leads to persistent deficiencies in regeneration 14 days after injury regardless of age. These results are found in new Figure 8, Results (last paragraph). The characterization of sublaminar Pax7 cells in 14 dpi P7S4-/- regenerating skeletal muscles was difficult to assess.

*5) The authors need to provide further insight into mechanism to support their conclusion that satellite cell expansion is disrupted with SMAD4 deletion. Inactivation of Smad4 during embryonic development leads to major defects in tongue muscle formation, which was attributed to impaired myogenic differentiation and fusion probably due to decreased expression of FGF4 and FGF6. Thus, the authors should investigate whether the adult phenotype similarly intersects with FGF signaling. To do so, they can use the existing data on tongue muscle development in the NCBI repository, then perform a targeted qPCR panel to identify the suspect FGF factors. The authors can add the suspect FGF exogenously to wt and Smad4 KO satellite cell cultures and ask if the defects they have observed are rescued. This will clarify whether Smad4 targets different pathways during development and adult life, which is a central tenet of the work.*

As requested we have examined the expression of relevant FGFRs and FGFs pertinent to myogenesis. The results can be found in new Figure 3—figure supplement 1, Results (subsection “Smad4 loss severely impairs SC clonal growth and proliferative potential”, third paragraph), and Discussion (second paragraph). Although similar to the Han et al., 2012 study we do find reduced expression of FGFR4 and FGF6. However, we also observe a reduction in *FGFR1*, no change in FGF1, and a significant induction of FGF2 expression. Although in the Results section (subsection “Smad4 loss severely impairs SC clonal growth and proliferative potential”, second paragraph) we only state 10% horse serum, DMEM, in the Materials and methods section under FACs purified SC/MP cultures we state FGF2 was used in our culture system. The latter is correct and we apologize for the confusion this may have caused and corrected the Results section accordingly. Although these data do provide similar results to the tongue study as it pertains to Smad4 loss in myogenic cells leading to decreased FGF6 and FGFR4 expression, the reduction in *FGFR1* and 4, and increased FGF2 expression are not congruent with the notion that FGF supplementation could somehow rescue deficiencies in proliferation. If anything, and consistent with the Han et al., 2012 study, such supplementation may promote terminal commitment. Therefore, based on our observations supplementation of individual FGFs was not conducted.